# CAR-T Cells in the Treatment of Ovarian Cancer: A Promising Cell Therapy

**DOI:** 10.3390/biom13030465

**Published:** 2023-03-02

**Authors:** Xi-Wen Zhang, Yi-Shi Wu, Tian-Min Xu, Man-Hua Cui

**Affiliations:** Department of Gynecology, The Second Hospital of Jilin University, Changchun 130000, China

**Keywords:** ovarian cancer, CAR, CAR-T, immunotherapy

## Abstract

Ovarian cancer (OC) is among the most common gynecologic malignancies with a poor prognosis and a high mortality rate. Most patients are diagnosed at an advanced stage (stage III or IV), with 5-year survival rates ranging from 25% to 47% worldwide. Surgical resection and first-line chemotherapy are the main treatment modalities for OC. However, patients usually relapse within a few years of initial treatment due to resistance to chemotherapy. Cell-based therapies, particularly adoptive T-cell therapy and chimeric antigen receptor T (CAR-T) cell therapy, represent an alternative immunotherapy approach with great potential for hematologic malignancies. However, the use of CAR-T-cell therapy for the treatment of OC is still associated with several difficulties. In this review, we comprehensively discuss recent innovations in CAR-T-cell engineering to improve clinical efficacy, as well as strategies to overcome the limitations of CAR-T-cell therapy in OC.

## 1. Introduction

OC is among the most common gynecologic malignancies with a poor prognosis and a high mortality rate [1]. Most patients are diagnosed at an advanced stage (stage III or IV), and even after aggressive surgical resection and chemotherapy, the 5-year survival rate is only 25–47% worldwide [1,2]. Therefore, there is a great need for new treatment strategies for these patients. Recently, immunotherapy has attracted great interest due to the improved understanding of the molecular basis of immune recognition and immune regulation in cancer cells [3]. Immunotherapy can boost the patient’s endogenous immunity to fight the tumor. The advent of immunotherapy has paved the way for a new era of cancer treatment that not only prolongs patients’ survival time but also enables them to make a full recovery.

In 1986, Rosenberg of the National Cancer Institute first proposed acquired T-cell therapy for cancer [4]. Tumor-infiltrating lymphocytes were surgically harvested, expanded in vitro, and then reinfused into patients. Complete remission was achieved in approximately 20% of patients with metastatic melanoma, and T cells were confirmed to have antitumor activity [5,6,7]. Tumor-infiltrating lymphocytes express natural T-cell receptors (TR) capable of recognizing antigens expressed by patient tumors in a major histocompatibility (MH)-restricted manner. To develop receptors that recognize tumor antigens independent of major histocompatibility complex (MHC) expression, researchers developed chimeric antigen receptors (CARs). Scientists took the lead in merging the variable regions of heavy and light chain monoclonal antibodies with the constant region of the TR and found that these synthetic receptors could recognize antigens and exert T-cell functions [8,9].

According to whether they are dependent on human leukocyte antigen (HLA), TR can be divided into two categories: one type is a specific HLA-dependent TR that can recognize tumor-expressed antigens in an MHC-restricted manner. The other type is a CAR-T, an HLA-independent receptor that can recognize tumor antigens independent of MHC expression. CAR-T therapy represents a major advance and is among the most promising cancer immunotherapies [10]. There are a variety of highly immunogenic antigens on the surface of ovarian tumors that can be used as targets for CAR-T cells [11]. However, the effect of CAR-T cells in OC is still associated with various difficulties [12]. Therefore, we described the production of CAR-T cells, summarized the difficulties of CAR-T-cell therapy in OC, and reviewed the strategies to overcome the dilemma of CAR-T-cell therapy.

## 2. CAR-T Cells Preparation

A CAR-T cell is formed by combining the single-chain variable region (scFv) of the monoclonal antibody and the T-cell coreceptors signaling region. For different tumor cell antigens, we can construct different monoclonal antibodies to replace the complementarity determining regions in the T-cell receptors (TCR), effectively bypassing the MHC restriction. The scFv in CAR directly transmits signals to T cells by specifically recognizing tumor-specific antigens, activating T cells and then secreting cytokines to kill tumor cells [13]. Compared to TCR, CAR antibodies can recognize a broader spectrum of corresponding tumor cell antigens, activate T cells in an MHC-unrestricted manner, and then exert highly efficient and specific antitumor effects [14,15].

Allogeneic CAR-T cells produced in the same batch can be used for multiple patients. The manufacturing process of CAR-T cells is as follows: ① T lymphocytes are taken from healthy individuals by leukapheresis. ② Transgenic or gene editing mediated by viral vectors is performed. Thus, gene-directed knock-in technology is used to recombine DNA, which is then inserted into CARs and other genes such as suicide genes or costimulatory receptors in lymphocytes. ③ αβ-T-cell receptor expression is eliminated on T cells (e.g., by gene editing-mediated knockdown of TCRα) and CD52. ④ Subsequently, anti-CD3/anti-CD28 beads and cytokines are used to expand T cells. ⑤ The αβ-TCR antibody magnetically removes the remaining αβ-TCR-positive cells. ⑥ Vials are filled with allogeneic CAR-T cells. ⑦ The product is stored, frozen, and shipped to the hospital as needed, as shown in Figure 1.

## 3. Antigen Targets

Tumor antigens are mainly divided into two categories: tumor-specific antigens and tumor-associated antigens. Tumor-specific antigens are specifically expressed on tumor cells and can, therefore, be targeted with fewer side effects. Tumor-associated antigens are expressed on both cancer cells and normal cells. They are less expressed on normal cells and have higher toxicity and more side effects when targeted.

Ovarian tumors often lack TSAs; thus, the target of immunotherapy is usually one or more tumor-associated antigens [16]. In the following subsections, we review the use of CAR-T therapy for OC, including erb-b2 receptor tyrosine kinase 2 (ERBB2), programmed cell death-ligand 1 (CD274), programmed cell death 1 (PDCD1), epithelial cell adhesion molecule (Ep-CAM), anti-Müllerian hormone receptor type 2 (AMHR2), annexin A2 (ANXA2), trophoblast glycoprotein (TPBG), folate receptor alpha (FOLR1), mesothelin (MSLN), mucin 16 (MUC16) and CD24 (Figure 2).

### 3.1. ERBB2

ERBB2, also known as human epidermal growth factor receptor 2 (HER2), is a proto-oncogene and plays an important role in the pathogenesis and clinical process of various tumors. In vitro and animal studies have clearly demonstrated that gene amplification and protein overexpression of HER2 plays a key role in tumorigenic transformation and development [17]. Moreover, HER2 is among the most studied tumor-associated antigens in cancer immunotherapy. Later studies have shown that amplification and overexpression of the HER2 gene is associated with OC [18], while protein expression is negative or very low in normal tissues. The monoclonal antibody trastuzumab has improved outcomes for patients with HER2+ breast cancer. However, a large number of HER2+ tumors do not respond to or develop resistance to trastuzumab-based therapy, and more effective HER2-targeted therapies are needed. Sun et al. [19] have developed a HER2-CAR-T-cell treatment and found that novel HER2-CAR-T cells can recognize and kill HER2+ OC cells. Currently, HER2-CAR-T-cell therapy has shown good therapeutic potential in the preclinical stage. However, HER2-CAR-T-cell therapy for OC is still in the clinical trial stage [1].

### 3.2. MSLN

MSLN, a group of glycoproteins anchored on the plasma membrane through the phosphatidylinositol region (GPI) [20], is normally expressed in pleura, peritoneum, pericardium and mesothelial cells. MSLN is highly expressed in 30% of OC. Because MSLN is often expressed on the surface of human normal tissues and its nonspecific toxicity is lower [21,22], researchers have designed a variety of therapeutic methods targeting MSLN antigens, such as antitoxins, antibody-based therapy, cancer vaccines and adoptive T-cell therapy. Preclinical and clinical experiments have shown good antitumor effects, suggesting that MSLN is a potential target [23]. Preclinical studies using MSLN-CAR-T cells in subcutaneous or in situ mouse models of mesothelioma, OC and lung transplantation have also been performed [21,22,24]. Recently, Zhang et al. [25] developed a MSLN-CAR containing the MSLN-scFv, the CD8 transmembrane domain, the costimulatory domains of CD28 and TNFRSF9 and the activation domain CD247. In vitro, the MSLN-CAR killed tumor cells from OC cell lines and showed that it can specifically kill MSLN-positive cells and release cytokines. In vivo, they investigated the effect of MSLN-CAR-T-cell therapy on xenografts of OC cell lines. Compared to the control group, the MSLN-CAR reduced the growth of MSLN-positive tumors, and T-cell and cytokine levels were significantly increased. Studies have shown that MSLN is a safe and feasible target for the treatment of CAR-T cells [22]. Therefore, MSLN-CAR-T cells are a promising and potential specific target in immunotherapy and may be a promising approach for OC.

### 3.3. MUC16

MUC16, also known as cancer antigen 125 (CA125), belongs to the mucin glycoprotein family and is the largest membrane-associated mucin, containing approximately 22,000 amino acids. It was originally detected by the mouse monoclonal antibody OC 125 (the 125th antibody produced against an OC cell line) and was given the name “cancer antigen 125”. MUC16 is used as a marker for endometrial, fallopian tube, non-small-cell lung, breast, pancreatic and gastrointestinal cancers. Thus, MUC16 plays a key role in promoting tumorigenesis and tumor proliferation, suggesting that it is a promising target for immunotherapy.

More than 80% of OCs are characterized by overexpression of MUC16, and high expression of MUC16 is an important indicator for early diagnosis of OC [26]. Studies have shown that MUC16-CAR-T cells have a specific killing effect on MUC16+ OC cells in vitro. Intravenous or intraperitoneal injection of MUC16-CAR-T cells can delay the progression of OC or completely clear up the tumors in mouse tumor models [27]. These studies also confirmed the research value of MUC16 as a potential target for the treatment of a class of OC cells.

### 3.4. EPCAM

EPCAM is overexpressed in OC cells and can be targeted by CAR-T cells. Fu et al. [28] developed a third-generation EPCAM-CAR containing EPCAM-scFv, the fragment that targets EPCAM, the CD8 transmembrane domain, the stimulatory domain of CD28 and TNFRSF9, and the CD247 activation domain. CAR is then transferred into T cells using lentiviruses. EPCAM-CAR-T cells were found to suppress OC tumor activity in vitro and in vivo. Thus, treatment with EPCAM-CAR-T cells may be a promising therapeutic approach for OC.

### 3.5. FOLR1

FOLR1 is highly expressed in OC, whereas it is scarcely expressed in normal tissues, making it an attractive target for immunotherapy. Overexpression of FOLR1 is significantly correlated with tumor malignancy and prognosis [29]. Zuo et al. [30] investigated FOLR1-specific CAR-modified cytokine-induced killer (CIK) cells and evaluated their antitumor immune effect on OC. They found that FOLR1-CAR-CIK cells enhanced antitumor immunity against FOLR1+ OC [30]. Furthermore, Kandalaf et al. [31] conducted a phase I clinical trial of adoptive transfer of folate receptor-α-targeted autologous T cells in the treatment of recurrent OC. They investigated the feasibility, safety, and activity of FOLR1-targeted CAR-T cells carrying the tumor necrosis factor receptor superfamily 9 (TNFRSF9) co-stimulatory domain in recurrent OC after lymphatic administration. It was found that treatment trials with CAR-T cells prolonged tumor recurrence time and overall survival in OC patients; however, dose-induced toxicity was not evaluated in this study. They suggested infusing large amounts of untransformed autologous peripheral blood lymphocytes two days after a low-dose infusion of CAR-T cells to suppress CAR-T-cell overexpansion in vivo and reduce toxicity.

### 3.6. AMHR2

AMHR2, a member of the transforming growth factor beta (TGFB) receptor family, is overexpressed by most ovarian and endometrial carcinomas and is not present in most normal tissues. The restricted tissue expression combined with the ability to transmit apoptotic signals to cancer cells makes AMHR2 an attractive target for tumor-targeted therapy. However, the development of clinical drugs targeting AMHR2 has always been challenging. Alba et al. [32] investigated a CAR targeting AMHR2 and found that AMHR2-CAR-T cells exhibited antigen-specific reactivity in vitro and eliminated tumors with high AMHR2 expression in vivo. AMHR2-CAR-T cells also recognized a number of human ovarian and endometrial cancer cell lines and lysed a number of patient tumor samples in vitro without killing normal human cells. The results demonstrate that AMHR2-CAR-T cells enable targeted therapy of OC and other gynecologic malignancies [32].

### 3.7. ANXA2

Leong et al. [33] developed an anti-membrane A2 CAR, CAR (2448), derived from an antibody active against epithelial OC cell lines. The interval length of CAR (2448) was optimized based on an in vitro cytotoxicity assay against OC cell lines and based on a real-time cytotoxicity assay. Longer-interval CAR (2448) LT cells showed significant effector activity by inducing an inflammatory cytokine release and cytotoxicity in OC cell lines. Furthermore, CAR (2448) L-BBzT cells improved survival and reduced tumor volume by 76.6% in an in vivo OC xenograft model. The preclinical study of Leong et al. [33] suggested that the CAR (2448) may be a new strategy for the treatment of OC.

### 3.8. TPBG

TPBG is a potential target for adoptive T-cell immunotherapy because it is highly expressed on the surface of various tumor cells and has low expression in normal tissues. Guo et al. [34] produced second-generation CAR-T cells specific for TPBG and showed that TPBG-CAR-T cells could induce lytic cytotoxicity and production of cytotoxic cytokines, including interferon gamma (IFNG), interleukin-2 (IL-2), and granulocyte-macrophage colony-stimulating factor. Moreover, adoptive transfer of TPBG-CAR-T cells significantly delayed tumor formation in peritoneal and subcutaneous animal model xenografts. These results demonstrate the potential therapeutic effect and feasibility of immunotherapy with TPBG-CAR-T cells and provide a theoretical basis for future clinical trials of OC immunotherapy [34].

### 3.9. CD24

Klapdor et al. [35] developed a novel anti-CD24-CAR, which was a third-generation codon-optimized CAR with a highly active scFv against CD24, namely, SWA11. Besides, they also equipped the human natural killer (NK) cell line NK-92 with an anti-CD24-CAR and an anti-CD19 control CAR using lentiviral transduction technology. The engineered NK-92 cells showed high cytotoxicity against CD24-positive OC cell lines (SKOV3 and OVCAR3). This effect was limited to CD24-expressing cells after the transduction of CD24-negative cell lines (A2780 and HEK-293T) with the CD24 transmembrane protein lentivirus. Moreover, to reduce the off-target effects in vivo, authors used a dual specific CAR targeting CD24 and MSLN antigens, and detected that the dual CAR was active in both CD24- and MSLN-expressing cells [35]. Therefore, the anti-CD24-CAR showed high cytotoxicity against OC cell lines and primary OC cells.

### 3.10. PDCD1 and CD274

Immunotherapy has demonstrated its great potential in the treatment of a variety of cancers, particularly through immune checkpoint inhibitors targeting PDCD1/ CD274 [36]. PDCD1 is mainly expressed by CD4+ and CD8+ T lymphocytes, whereas its ligand, CD274, is widely expressed in various cell types including activated lymphocytes, fibroblasts, tumor-associated macrophages and tumor cells [36]. A study showed that nearly two-thirds of ovarian tumors had a modest expression of CD274 mainly on immune cells rather than tumor cells, which was associated with the worst prognosis [36].

Several approaches have been proposed to reduce the sensitivity of T cells to immunosuppression, such as engineering CAR-T cells with PDCD1 interference or CARs expressing PDCD1-CD28 and using the cytoplasmic portion of CD28 to convert inhibitor signals into activator signals [37,38]. Recently, it was reported that a modified CAR-T-cell can secrete a single-chain variable fragment that blocks PDCD1 with enhanced antitumor activity [39].

Fang et al. [40] reported a patient with refractory OC who relapsed after several chemotherapies. The patients received a full-length antibody against PDCD1 (αPDCD1) and autologous T cells containing sequences encoding MSLN-specific single-chain variable fragments. The modified T cells are called αPDCD1/MESO-CAR-T cells. After infusion, the number of CAR-T cells was detected in the blood and the secretion of PDCD1 antibody was increased. Using multimodal tumor tracking, an MRI of the liver showed that the mean diameter of metastatic nodules decreased from 71.3 mm to 39.1 mm after 2 months. The patient achieved a partial response and survived more than 17 months. The patient’s IL-6 scores improved 2–4-fold from baseline after treatment, and side effects were minimal, with only grade 1 hypertension and mild fatigue. Therefore, αPDCD1/MESO-CAR-T-cell therapy showed potential therapeutic effect in advanced refractory OC [40]. Moreover, Li et al. [41] constructed tandem CAR-T cells, referred to as PDCD1/MUC16-CAR-T cells. PDCD1/MUC16-CAR-T cells showed a strong ability to kill OVCAR-3 cells and released a large number of cytokines, such as IL-2, IFNG and tumor necrosis factor (TNF).

## 4. Challenges for CAR-T-Cell Therapy

Currently, CAR-T-cell therapy for OC still faces many challenges, including off-target effects, tumor antigen escape, heterogeneity of ovarian tumors and tumor microenvironment (TME) of immunosuppressive tumor cells (Figure 3).

### 4.1. Off-Target Effects

CAR-T cells are effective in recognizing and destroying tumor cells. However, when there are target cells in both tumor and nontumor tissues, CAR-T cells can remove both normal cells and tumor cells, which is an off-target effect. CAR-T cells targeting the CD19 protein can recognize and destroy B-cell lymphomas and leukemia cells, while also destroying all of a patient’s normal CD19+ B cells. Thus, if the target protein is expressed in normal epithelial cells, this can have serious consequences.

### 4.2. Tumor Antigen Escape

Target antigen expression is greater in ovarian tumors than in hematologic tumors, so tumor antigen escape, including antigen loss or downregulation, could significantly compromise therapeutic efficacy. Given the extensive clinical experience with CD19-CARs in pediatric B-ALL, most of the data on the mechanisms of antigen loss come from studies using patient samples from these trials [42,43,44]. Loss of CD19 antigen occurs by two distinct mechanisms: antigen escape or lineage switching [45]. In antigen escape, after CD19-CAR remission, patients relapse with a phenotypically similar disease that lacks surface-expressed CD19 molecules capable of binding to anti-CD19 antibodies in the CAR (Figure 4). Based on the mechanism of tumor antigen escape, scholars have explored numerous methods to avoid immune escape, including bicistronic CAR-T cells, tandem CAR-T cells, co-administered CAR-T cells, and co-transduced CAR-T cells (Figure 5).

### 4.3. Heterogeneity of Ovarian Tumors

OC, unlike blood tumors, is a solid tumor. The expression of target antigens on the surface of different pathological types of tumor cells is not completely uniform, making it difficult for a single type of CAR-T cell to kill all tumor cells, leading to problems in selecting tumor-specific target antigens. This difficulty also limits the specificity and wide application of CAR-T cells.

### 4.4. Immune Checkpoint

Tumor-infiltrating T cells are functionally resistant to the tumor microenvironment and exhibit suppressed effector functions compared to normal effector T cells [46]. They express inhibitory receptors, such as PDCD1, TIM-3, TIGIT, LAG-3 and cytotoxic T-lymphocyte-associated protein 4 (CTLA4) [47,48,49], which induce T-cell exhaustion and prevent T-cell activation; however, cancer cells utilize these receptors to evade T-cell attacks [50,51,52]. PDCD1 regulates T-cell activity by binding to CD274 [53] or PD-L2 [54] on the surface of tumor cells [55]. PDCD1 following binding by its ligands CD274 and PDCD1LG2 inhibits the proliferation of T cells as well as the production of IFN, TNF and IL-2 and also limits the survival time of T cells [56]. Similar to PDCD1, CTLA4 following binding by its ligands CD80 and CD86 and inhibits the activity of CAR-T cells directly at the TCR immune synapse [57,58]. Moreover, the high expression of immune checkpoint proteins on CAR-T cells has been associated with decreased tumor killing ability, loss of tumor immune response and poor prognosis [59]. In addition to PDCD1 and CTLA4, several co-inhibitory receptors such as TIM-3, LAG-3 and TIGIT are also expressed on depleted T cells [60]. Therefore, blocking the immune checkpoint may further enhance CAR-T-cell function.

### 4.5. T-Cell Exhaustion

There is an unfavorable phenomenon that occurs when CAR-T cells are used to treat ovarian tumors: T cells that invade ovarian tumors may stop killing due to “T-cell exhaustion”, resulting in CAR-T cells being ineffective. Studies have shown that a family of transcription factors called nuclear receptor subfamily 4 group A number 1 (NR4A1), which plays an important role in “T-cell exhaustion”. Injection of NR4A1-deficient CAR-T cells into mice bearing a tumor can significantly inhibit tumor growth [61].

### 4.6. Efficiency of CAR-T Cells after Reaching the Tumor Site Is Difficult to Guarantee

To reach the tumor cell site, CAR-T cells must penetrate the extracellular matrix and then overcome unfavorable factors such as the immunosuppressive microenvironment. When CAR-T cells reach the tumor cells, their number and activity are reduced, and their efficiency in killing tumor cells is also greatly reduced. Therefore, improving the homing ability of CAR-T cells may help to improve the effect of CAR-T cells.

### 4.7. TME of Tumor Immunosuppressive Cells

#### 4.7.1. Immunosuppressive Cells

In ovarian tumors, the suppression of the CAR-T-cell antitumor immune response is mediated by immunosuppressive cells, including regulatory T cells, myeloid-derived suppressor cells (MDSCs) and TAMs in the TME. These cells release cytokines such as TGFB and IL-10 to inhibit the action of CAR-T cells [62]. MDSCs are representative immunosuppressive cells that not only inhibit the antitumor activity of CAR-T cells but also promote tumor angiogenesis and increase tumor invasiveness. MDSCs also inhibit the proliferation and function of T cells through arginase, iNOS, TGFB and IL-10 [63]. Moreover, MDSCs induce the production of reactive oxygen species (ROS), forming a potent immunosuppressive microenvironment. Therefore, the inhibition or depletion of MDSCs may enhance antitumor immune responses in the TME. Most tumor cells induce immunosuppressive M2 macrophages, but not immunostimulatory M1 macrophages. TAMs promote tumor cell proliferation, activation and metastasis by expressing a variety of immunosuppressive soluble factors [64].

#### 4.7.2. Reactive Oxygen Species (ROS)

The TME of ovarian tumors inhibits the function of infiltrating T cells by several mechanisms, such as the production of ROS [65,66]. Infiltrating T cells are often subjected to significant oxidative stress, which impairs their antitumor activity. The accumulation of ROS in the TME is induced by MDSCs. The oxidative stress released by MDSCs alters TCR and CD8 or downregulates CD3 expression, thereby suppressing antigen-specific responses [67] and effect T-cell activity [68]. In the TME, ROS is overproduced due to an imbalance between ROS production and excretion. Oxidative stress is known to cause cellular damage through mutations, especially mutations in genes related to the cell cycle and the tumor suppressor gene p53, leading to tumorigenesis and tumor growth [69]. Ligtenberg et al. [70] engineered the co-expression of H_2_O_2_ and tumor-specific CAR to enhance the adaptation of antitumor T cells to ROS, and found that these CAR-T cells exhibited increased H_2_O_2_ enzyme levels and decreased ROS accumulation.

#### 4.7.3. Metabolites

T-cell-mediated antitumor responses require a balance of nutrients and energy to promote T-cell expansion and function. However, tumor cells have high metabolic demands, leading to the inhibition of CAR-T-cell function in the TME. In mouse tumor models, reducing metabolites can inhibit the MDSC-mediated suppression of T cells and enhance antitumor activity [71]. Therefore, it is important to control metabolites in the TME. Lactate, among the metabolites of tumor cells, inhibits T-cell proliferation, activation, cytokine production and cytotoxicity [72]. Other metabolites of tumor cells, including adenosine [73], indoleamine-2,3-dioxygenase [74], arginase-1 [75] and L-arginine [76], are amino-acid-degrading enzymes commonly expressed at TME and also inhibit T-cell function [77,78,79]. Baumann et al. [71] reported that MDSC-dependent metabolites are transferred from cell to cell until they reach the cell membrane of T cells through the toxic metabolite methylglyoxal (MG), which restricts the antitumor response of CD8+ T cells. The MG-responsible factors and MG-related signaling pathways in T cells are unclear. Previous studies have shown that MG is associated with oxidative stress responses involving the nuclear factor erythroid 2-related factor 2 (NRF2) [80] and hippo signaling pathways [81]. In summary, an optimal balance between T-cell and tumor cell metabolism is critical for tumor control. The modulation of metabolites can improve the function of CAR-T cells, including promoting the formation of central memory-like cells and reducing T-cell death [82].

#### 4.7.4. Cytokines

Tumor-derived cytokines are associated with inflammatory responses at the tumor site and may impair the antitumor response of CAR-T cells in a way similar to how immunosuppressive cytokines suppress the antitumor response of CAR-T cells in ovarian tumors [83]. At the tumor site, immunosuppressive cells produce immunosuppressive cytokines, such as TGFB, IL-10 and IL-6 [84,85], playing a key role in reducing the antitumor function of T cells. The expression of IL-10 inhibits the activation of T cells and NK cells, leading to the proliferation and activation of tumor cells [86]. Among the most critical suppressive, tumor-associated cytokines is TGFB [87,88]. TGFB can inhibit CD8+ T-cell function and induce regulatory T-cell maturation; thus, TGFB inhibition with antibodies can improve CD8+ T-cell function and enhance CAR-T-cell persistence [89].

#### 4.7.5. PH Value

Oxidative stress is clearly associated with an acidic microenvironment. This acidic condition promotes tumor development and metastasis. Many areas in tumor mass are acidic. In ovarian tumor cells, glucose metabolism is increased, leading to high production of lactate and H^+^ [72,90]. The acids produced in the tumor cells are transported out of the cell, making the extracellular space acidic. Tumor cells are well adapted to an acidic microenvironment. In vitro studies have shown that the optimal proliferation of tumor cells occurs at a pH of 6.8, rather than the pH of 7.3 required by healthy cells [91]. Low pH triggers the formation of fibrous structures around the tumor that limit the normal motility and migration of T cells [92] and result in T cells being unable to respond to tumor cells [93].

#### 4.7.6. Hypoxia

A harsh TME causes hypoxic (O_2_) stress [94], which can lead to dysfunction of T cells infiltrating the tumor, resulting in tumor resistance to drug and even tumor progression [95,96]. Recent studies have reported that designing CAR-T cells with oxygen-sensitive factors such as HIF-1β leads to the promotion of memory-related metabolic pathways, such as fatty acid oxidation, and improved functions in harsh microenvironments [78,97,98]. The expression of catalase-enhanced CAR-T cells helped maintain their antitumor activity under H_2_O_2_-induced oxidative stress [70].

### 4.8. Cytokine Release Syndrome (CRS)

CRS is a systemic inflammatory response that can be triggered by a variety of factors including infections and certain drugs [99]. CAR-T cells are overactivated and cannot be effectively controlled, resulting in a syndrome of cytokine release [1]. However, in clinical trials, the rapid increase in cytokines can lead to a “CRS” in which patients present with mild symptoms such as fever, fatigue, headache, rash, arthralgia and myalgia and severe symptoms such as a systemic inflammatory response, shock, vascular leakage, diffuse intravascular coagulation, multi-organ failure and even death. Therefore, cytokines are the “double-edged sword” of immunotherapy [99].

## 5. Strategies to Enhance the Efficacy of CAR-T Cells

The immunosuppressive microenvironment and off-target effects have severely compromised the therapeutic effect of CAR-T cells in OC. To improve the antitumor efficacy of CAR-T-cell therapy, researchers have made many attempts to improve the current situation. In this section, we review the various methods currently being tested in clinical trials (Figure 6).

### 5.1. Armored CAR-T

Armored CAR-T cells are CAR-T cells modified to secrete proinflammatory cytokines to protect CAR-T cells from TME. Armored CAR-T cells modified to secrete proinflammatory IL-12 cytokines have shown encouraging results in several preclinical studies [100,101,102,103]. In addition, T cells engineered to secrete cytokines such as IL-15 [104], IL-18 [105] and IL-21 [106] have been shown to enhance the proinflammatory effects. Compared to CAR-expressing T cells alone, the use of armored CAR-T cells results in enhanced antitumor efficacy due to the increased suppressive function mediated by CAR-T cells and immunosuppression of the transplanted T cells in the TME. Cell resistance is increased, including regulatory T cells (Treg) and myeloid-derived suppressor cells (MDSCs) [101,103].

### 5.2. Dual Receptors and Chemokine Receptors

To improve the persistence of CAR-T cells, some scientists have engineered T cells that can express two receptors: one to recognize tumor-associated antigens and the other to provide cytokines to the T cells to maintain T-cell stability [107]. For example, the tumor-associated antigen recognition receptor T1E28z consists of a second-generation CAR signaling domain (CD28/CD247) linked to T1E, a chimeric polypeptide that binds both to HER1 homoplasmically and to HER2/HER3 heteroplasmically [108]. This signaling receptor is a chimeric cytokine receptor (4αβ) in which the ectodomain of the IL-4 receptor is linked to the β-chain subunit shared by the IL-2 and IL-15 receptors, allowing IL-2/IL-15 to generate similar proinflammatory stimulatory signals. This cytokine plays a role in the immune microenvironment of many tumor types [107].

### 5.3. NK Cell Receptors

NK cells express receptors that can distinguish tumor cells from nonmalignant cells. These receptors are used as antigen-recognition domains by CARs to enhance tumor recognition by T cells. Among these receptors is NKG2-D, and cognate ligands include stress-induced proteins that are overexpressed in many tumor types [109]. By linking the extracellular domain of NKG2-D to the intracellular T-cell signaling domain, NKG2-D ligands can activate T cells, which is an approach that has been shown to have antitumor activity in preclinical studies [110].

### 5.4. Improving CAR-T-Cell Persistence

Studies have confirmed that one possible reason for the failure of CAR-T-cell therapy is decreased CAR-T-cell persistence due to immune recognition of CAR-derived foreign peptides and the subsequent immune-mediated destruction of the modified T cells [52,111]. Ajina and Maher et al. [112], Kim et al. [113], and Scott et al. [114] indicated that infection with oncolytic viruses can enhance the entry and motility of CAR-T cells and alleviate or reverse local immunosuppression, which increases CAR-T-cell persistence. Moreover, converting the immunosuppressive environment to an activating environment, e.g., by inducing the tumor to release IL-4, has an inhibitory effect on T cells. A reverse cytokine receptor on T cells with the extracellular domain of IL-4 and the intracellular domain of IL-7 was developed. The results showed that reverse cytokine receptors can convert inhibitory signals from antigen-specific CAR-T cells of prostate stem cells into proliferative signals from T cells and cause enhanced antitumor immunity in prostate cancer models [115]. Another team took a similar approach and constructed a chimeric PDCD1 receptor with a truncated PDCD1 extracellular domain and the transmembrane and internal domains of CD28, converting PDCD1 signaling into stimulatory signaling [116]. Moreover, replacement of IL-2 with IL-7 or IL-15 in the in vitro expansion of memory T cells improves the phenotype of memory T cells, reduces failure markers, improves cell proliferation, and has an anti-apoptotic function [117,118].

### 5.5. Tumor Vessel-Targeted CAR-T Cells

CAR-T cells targeting tumor blood vessels are being investigated for the treatment of metastatic OC. Angiogenic growth factors such as VEGF present in TME and overexpression of their receptors in tumor cells are associated with poor prognosis and tumor metastasis [119]. VEGFR-2 was found to be overexpressed on tumor stromal cells and some other tumor cells [120]. Targeting VEGFR-2 with the CAR can switch the killing effect of T cells to an effect that targets tumor stromal cells while protecting normal tissues [121]. Local administration of IL-12 with VEGFR-2-CAR-T cells resulted in increased survival in mice with subcutaneous tumors [96].

### 5.6. Promoting Homing and Aggregation of CAR-T Cells in Tumor Tissue

Ovarian tumor tissue is rich in blood vessels, fibroblasts, and myeloid cells. These features protect tumor cells and prevent infiltration of CAR-T cells [122]. Among the preferred ways to solve this problem is the local administration of CAR-T cells. This can be achieved in most tumor models by intratumoral drug delivery. In malignant pleural cancer, intrapleural administration showed greater activity and persistence compared with intravenous administration of CAR-T cells [123]. Local delivery by intracranial/intracerebroventricular administration has also been tested with good results in head and neck cancer and glioma models [123,124,125]. Furthermore, CAR-T cells do not need to be injected into the lesion. The cells can first be taken up by the biopolymer along with other coactivating molecules. Then, the biopolymer scaffold can be placed near the tumor mass, and the CAR-T cells are finally released at the optimal concentration and at a constant rate [126].

Chemokines are important molecules that promote the infiltration of T cells into tumor foci. They bind to the corresponding receptors on T cells, leading to activation of intracellular signaling mechanisms and resulting in cell migration and high concentrations of chemokines [127]. In vivo, a variety of cells express chemokines, including stromal cells, cells of the innate immune system, and even tumor cells [128,129,130]. The chemokines expressed in the TME often do not match the receptors expressed in T cells. This mismatch of chemokine and chemokine receptors is among the best-studied and exploited mechanisms for insufficient T-cell infiltration and subsequent immune evasion of the tumor [131].

Homologous chemokine receptors on CAR-T cells have been developed to improve antitumor efficacy. These experiments began in 2002 when Kershaw et al. [132] engineered a CXCR2 receptor on T cells for its cognate cytokine CXCL1, resulting in improved chemotactic behavior of T cells. In another study, C-C motif chemokine receptor 2 (CCR2) on CAR-T cells was modified to target the cognate chemokine CCL2, which is highly expressed in malignant pleural mesothelioma. The engineered CAR-T cells showed increased migration and 12.5-fold cytotoxicity compared to non-transduced CAR-T cells [133]. CCR2b transduction of GD2, a glycolipid antigen expressed by CAR-T cells, was also studied in a neuroblastoma model. A previous study showed a 10% enhancement of homing in vitro and increased antitumor activity in vivo [134].

In 2017, a study was published on the key protein RUNX3, which induces T cells to leave lymphoid tissues and accumulate in peripheral and tumor tissues [135]. RUNX3 can reprogram T-cell genes to promote the expression of genes related to T-cell settlement in tissues, thus promoting the accumulation of CAR-T cells in tumor tissues. The settlement and aggregation of CAR-T cells in tumor tissues also depends on the recognition and binding of chemokine receptors (CCRs) on the surface of T cells with the corresponding chemokines on the surface of tumor cells. The accumulation of CAR-T cells in tumor tissues can be promoted by the overexpression of CCRs from T cells (e.g., CCR2b and C-C motif chemokine receptor 4) or by the overexpression of chemokines from tumor cells (e.g., RANTES and IL-15) [136]. To enhance the homing ability of CAR-T cells, they can also be induced to overexpress heparinase. By local injection into the tumor, CAR-T cells can directly enter the tumor tissue and enhance the direct killing of tumor cells [137]. Another important recent approach is the use of mesenchymal stem cells (MSCs), which can colonize tumor tissue to modulate the tumor’s immunosuppressive microenvironment. MSCs engineered to produce both IL-7 and IL-12 are critical for T-cell survival and protective responses, transforming the suppressive TME into a stimulatory one.

### 5.7. Multiantigen-Targeted CAR-T Cells

CAR-T-cell development is challenging because of the unique antigens expressed on tumor cells. In ovarian tumors, CAR-T cells target tumor-associated antigens that are also present in normal cells [138]. In addition, tumor resistance to CAR-T cells may be caused by antigen escape, including downregulation, reduction and/or elimination of antigens [139]. To overcome antigen escape and mitigate on-target and off-tumor toxicities, CARs targeting multiple antigens are being tested. These include pooled CAR-T cells, which are mixtures of two engineered T-cell lines, each expressing a different antigen-specific CAR; dual CAR-T cells, which are two individual CARs targeting different antigens in a single T cell; Tandem CAR-T cells, in which two different antigen-binding domains are linked in tandem to a single CAR; and trivalent CAR-T cells, which are three CARs targeting specific antigen molecules in a single engineered T cell.

To achieve high efficacy and low toxicity with the multiantigen targeting strategy, Boolean logic was applied to “gate” the activity of CAR-T cells based on multiantigen targeting. For example, T cells with two independent CAR molecules or a pool mixed with different specific CAR-T cells can only use the “AND” logic gate, which allows CAR-T cells to target the tumor simultaneously in the presence of both target antigens. The “AND” logic gate provides a better balance in the relationship between antitumor efficacy and on-target and off-tumor toxicities. The “OR” logic gate CAR-T cells can be activated in the presence of either antigen, expanding antigen coverage and reducing antigen escape. However, a “OR” logic gate could potentially have more severe on-target and off-tumor toxicities. In addition, the “NOT” logic gate can also be used to allow the engineered T cells to distinguish target cells from non-target cells and avoid attacking normal tissues, thereby achieving the safety of CAR-T cells [140].

### 5.8. Combination with Checkpoint Blockers

CAR-T cells are influenced by inhibitory immune checkpoint signals in the TME, such as programmed cell ligand PDCD1 or CTLA4 ligands. These inhibitory receptor–ligand interactions can be blocked with monoclonal antibodies to remove the inhibition of T cells. Although this approach has shown encouraging results in some malignancies, the success of these therapies clearly depends on a pre-existing, endogenous, tumor-specific T-cell response [141,142,143]. Indeed, CAR-T cells are susceptible to PDCD1-mediated inhibition. Therefore, CAR-T-cell therapy in combination with monoclonal antibody immune checkpoint inhibitors is promising for protecting the function of CAR-T cells in TME [144].

In preclinical studies, John et al. [144], Liu et al. [116] and Cogdill et al. [145] found that CAR-T-cell therapy and a PDCD1 blockade were highly synergistic and could achieve long-term survival without causing side effects. Currently, there are few studies on the use of CAR-T in combination with PDCD1 in the treatment of OC. We have studied other solid tumors such as lung cancer and prostate cancer treated with CAR-T in combination with PDCD1. To overcome the immunosuppressive microenvironment, Li et al. [146] engineered T cells to secrete checkpoint inhibitors targeting PDCD1 (CAR.αPD1-T) and investigated their therapeutic efficacy in a human lung cancer xenograft mouse model. It was found that anti PDCD1 can enhance the antitumor activity of CAR-T cells and prolong overall survival. Serganova et al. [147] found that prostate-specific membrane antigen-specific CAR-T-cell therapy alone was ineffective in treating prostate cancer, while CAR-T-cell therapy combined with PDCD1 blockade resulted in better short-term outcomes in some patients [147]. Moreover, Gargett et al. [148] reported that the combination of PDCD1 checkpoint inhibitors with CAR-T-cell therapy may help improve the efficacy and persistence of CAR-T cells in patients (Figure 7).

## 6. CAR-T Cells in Combination with Radiotherapy or Chemotherapy

To date, CAR-T cells still face many challenges for OC [149]. Therefore, it is necessary to improve the efficacy of CAR-T-cell therapy in ovarian tumors [150].

### 6.1. CAR-T-Cell Therapy in Combination with Chemotherapy

Both preclinical and clinical studies have shown that CAR-T-cell therapy and chemotherapy alone are insufficient to eradicate large ovarian tumors, resulting in refractory tumors or subsequent recurrence [151]. Some scholars have suggested that the combined use of CAR-T-cell therapy and chemotherapy may improve the therapeutic effect [152,153].

#### 6.1.1. Chemotherapy Improves the Efficacy of CAR-T Therapy

Chemotherapeutic agents, including cyclophosphamide, doxorubicin, oxaliplatin, fluorouracil and gemcitabine, not only reduce the tumor burden but also have significant immunomodulatory effects [154,155,156]. The combination of immunotherapy and chemotherapy can achieve better therapeutic effect than monotherapy [152]. As shown in Figure 8, we summarize the pathway of an immune response induced by a chemotherapeutic agent to improve the effect of CAR-T-cell therapy, and then analyze the feasibility of using CAR-T cells in combination with chemotherapy to treat ovarian tumors.

#### 6.1.2. Chemotherapy Enhances the Sensitivity of Tumor Cells to Immunotherapy

Mannose-6-phosphate receptors on the surface of tumor cells are upregulated after treatment with certain chemotherapeutic agents, making it easier for granzyme B released by cytotoxic T lymphocytes (CTLs) to penetrate tumor cells and self-regulate. A phagocytosis-dependent manner sensitizes tumor cells to immunotherapy [157,158,159]. In addition, a preclinical study using ErbB-targeted T cells in combination with carboplatin has shown that treatment with the low-dose chemotherapeutic agent carboplatin increases the susceptibility of tumor cells to specific ErbB CAR-T cell-mediated cytotoxicity and enhances antitumor efficacy [160,161]. Although the mechanism of increased susceptibility to immunotherapy after treatment with chemotherapeutic agents is not fully understood, improved efficacy with combination therapy has also been observed in other studies [162].

#### 6.1.3. Chemotherapy Improves Tumor Antigen Expression and Recognition

Chemotherapeutic drugs such as taxanes (docetaxel and paclitaxel) and vinca alkaloids (vinorelbine and vinblastine) can enhance tumor cell recognition by increasing calreticulin exposure and killing tumor cells. This releases a large number of tumor antigens, increasing the likelihood that they will be recognized for killing [163]. In addition, studies have shown that some chemotherapeutic agents can increase tumor antigen expression, mainly in the following ways: First, autophagy induced by some chemotherapeutic agents stimulates the release of adenosine triphosphate (ATP) from tumor cells, which increases the recruitment of dendritic cells and T lymphocytes to infiltrate the tumor bed for tumor antigen expression [164]. Second, chemotherapeutically treated dying tumor cells release damage-associated molecular patterns such as hypermobility group box 1, which can be recognized by Toll-like receptor 4 (TLR4) to promote dendritic cell maturation and activation and enhance the antitumor T-cell response [165]. Third, chemotherapeutic agents stimulate tumor cells or stromal cells to produce endogenous type I interferon, and an increase in exogenous type I interferon can activate dendritic cells, induce the cross-stimulation of T cells and lead to tumor control [152,156].

#### 6.1.4. Chemotherapy Relieves Immunosuppressive Cells

Chemotherapeutic drugs (doxorubicin, fluorouracil, gemcitabine, cyclophosphamide, and docetaxel) can selectively suppress immunosuppressive cells (regulatory T cells and myeloid-derived suppressor cells) to improve the antitumor effect of treatment [166]. Certain immunosuppressive cells are more sensitive to certain chemotherapeutic agents than T cells, and low-dose chemotherapeutic agents are not harmful to ATC [167].

#### 6.1.5. Chemotherapy Improves CAR-T-Cell Durability

Several preclinical and clinical studies have shown that autoimmunity is suppressed after repeated chemotherapy, which is a very serious side effect [168]. After autoimmunity is suppressed, this may increase the efficacy of transferring ATC in cancer patients [153]. Direct CAR-T-cell therapy without chemotherapy has a short duration and a small therapeutic effect [169]. However, after chemotherapy, CAR-T-cell therapy was performed, and it was found that the persistence of CAR-T cells in the body was prolonged and the treatment effect was improved. This is because chemotherapy can suppress autoimmunity and remove suppressor cells, eliminating the potential immunogenicity of CAR-T cells [170].

#### 6.1.6. CAR-T Cells Improve the Treatment Effect of Chemotherapy

The innate and adaptive immune systems can largely contribute to the therapeutic effect of chemotherapy on cancer [171]. Wang et al. [172] showed that effector T cells abolish stroma-mediated chemo-resistance in OC. Fibroblasts release glutathione and cysteine, which contribute to chemotherapy resistance. T cells are able to release IFNG into fibroblasts via the activator Janus kinase 1/signal transduction pathway 1, which alters glutathione and cysteine metabolism and affects chemo-resistance. Furthermore, a growing number of preclinical and clinical studies have shown that autoimmunity is suppressed after several cycles of chemotherapy, which is a fatal side effect [168]. Therefore, CAR-T cells may enhance the therapeutic effect of chemotherapy.

### 6.2. CAR-T-Cell Therapy in Combination with Radiotherapy

#### 6.2.1. Radiotherapy Improves the Efficacy of CAR-T Therapy

Some researchers have found that radiotherapy can not only destroy tumor cells but also stimulate tumor-specific immunity to improve local and distant tumor control [173]. This finding proves that radiotherapy can improve the therapeutic effect of CAR-T cells in OC, as shown in Figure 9.

#### 6.2.2. Radiation Increases the Sensitivity of Tumor Cells to Tumor-Specific Cytotoxic Lymphocytes

Studies have shown that the local irradiation of tumors can increase the expression of MHC class I molecules and tumor-specific antigens on irradiated tumor cells. This makes them more receptive to tumor-specific cytotoxic lymphocytes, which improves the efficacy of adoptive immunotherapy with cytotoxic lymphocytes (CTLs) [174].

#### 6.2.3. Radiotherapy Improves CAR-T-Cell Delivery

The delivery and infiltration of effector T cells in ovarian tumors are critical steps for successful antitumor activity [175]. Studies have shown that the release of IFNG and DAMP is increased after irradiation, thereby attracting immune effector cells to the tumor microenvironment (TME), increasing the transport capacity of immune effector cells and creating a TME favorable for T-cell infiltration [176]. Consistent with this finding, it has been proposed that local irradiation can induce the expression of certain chemokines, including C-X-C motif chemokine ligands (CXCL) 9, 10 and 16, to promote T-cell settlement in the TME and enhanced infiltration of immune effector cells [177,178]. In addition, local irradiation also contributes to greater infiltration of lymphocytes into the tumor by reversing the non-adherent phenotype of the tumor endothelium [179].

#### 6.2.4. Radiation Therapy Improves Tumor Antigen Presentation

A growing number of studies have shown that radiotherapy can increase tumor antigen presentation [180]. Radiation therapy induces apoptosis and necrosis in tumor cells, causing them to release danger signals, including high mobility group box 1 (HMGB1) [181]. Subsequently, danger signals and tumor antigens may trigger type I IFNs in the tumor microenvironment as a link between innate responses and adaptive immunity [182,183]. This interplay between innate responses and adaptive immunity plays a critical role in promoting the maturation and activation of dendritic cells for enhanced presentation of tumor antigens [184,185,186].

#### 6.2.5. CAR-T Cells Improve the Treatment Effect of Radiotherapy

It has been suggested that radiotherapy is often associated with local or distant tumor recurrence and that the response to radiotherapy depends, in part, on the tumor microenvironment and the local immune system, especially in T cells [187]. Studies have shown that CD8+ T cells and their cytokines play an important role in maintaining control of irradiated solid tumors to reduce recurrence and metastasis [175,188,189].

Furthermore, after localized radiotherapy, CTLs can not only be attracted to the irradiated tissue to induce a local response, but also suppress distant tumors, a phenomenon known as the distant effect [190]. This has transformed radiotherapy from a regional antitumor therapy to one that can target distant metastases. In addition, CAR-T-cell therapy can enhance T-cell function. Therefore, by combining radiotherapy with CAR-T-cell therapy, it should be possible to further enhance the antitumor effect [191]. Researchers speculate that CAR-T-cell therapy in combination with radiotherapy may play a synergistic role in the treatment of OC [192].

### 6.3. CAR-T-Cell Therapy in Combination with Radiation and Chemotherapy

Chemoradiation (CRT) plays an important role in the treatment of OC. However, Yovino and Grossman [193] point out that CRT can cause therapy-related toxicity, including effects on host immunity, such as lymphopenia. In addition, many studies have shown that treatment of OC with CRT depletes T cells, resulting in the elimination of a subset of immune cells [194,195]. This may reduce the antitumor capacity of CRT and lead to an increased incidence of tumor metastasis and recurrence [196,197]. In particular, T cells are critical for mediating cellular immunity against tumor cells [198,199,200]. Therefore, injecting ATC to enhance the antitumor effect of radiotherapy and chemotherapy is a novel idea to prevent tumor metastasis and recurrence. Unfortunately, because the use of CAR-T-cell therapy in OC is still in its infancy, the combination of chemoradiotherapy and CAR-T-cell therapy in OC has not been explored as thoroughly. Although the clinical effect and mechanism of action of CAR-T-cell therapy in combination with CRT in OC are unclear, it remains a promising therapeutic approach. The immune system plays a key role in tumor control, and low absolute lymphocyte counts during treatment have been associated with worse clinical outcomes [201,202,203]. Buka et al. [204] indicated that CRT in combination with CAR-T therapy may achieve better outcomes in the treatment of solid tumors than CRT alone due to the increased T-cell density after CRT. The median survival rate of patients receiving CRT in combination with CAR-T-cell therapy is 2.5 times that of CRT alone. Zitvogel et al. [205], Aranda et al. [206] and Sathyanarayanan et al. [207] indicated that CRT may promote the antitumor efficacy of CAR-T-cell therapy because some CRT regimens can stimulate T cells.

## 7. Conclusions

OC is among the most common gynecologic malignancies worldwide. CAR-T-cell therapy offers a new direction for the treatment of OC. Several antigenic targets, including ERBB2, MSLN, MUC16, EPCAM, FOLR1, AMHR2, ANXA2, TPBG, CD24, PDCD1 and CD274, have been investigated and used for the treatment of OC. However, there are still many challenges in CAR-T-cell therapy for OC, such as the lack of tumor-specific targets, tumor antigen escape, TME of tumor immunosuppressive cells, and off-target effects. Accordingly, many strategies have been proposed to improve the efficacy of CAR-T cells. These include engineering armored CAR-T cells, improving the persistence of CAR-T cells, developing CAR-T cells to target tumor vessels, promoting the settlement and aggregation of CAR-T cells in tumor tissues, using CAR-T cells that target multiple antigens, and combining them with checkpoint blockers and chemotherapy. In short, CAR-T-cell therapy for the treatment of OC is an extremely promising treatment modality.

## Figures and Tables

**Figure 1 biomolecules-13-00465-f001:**
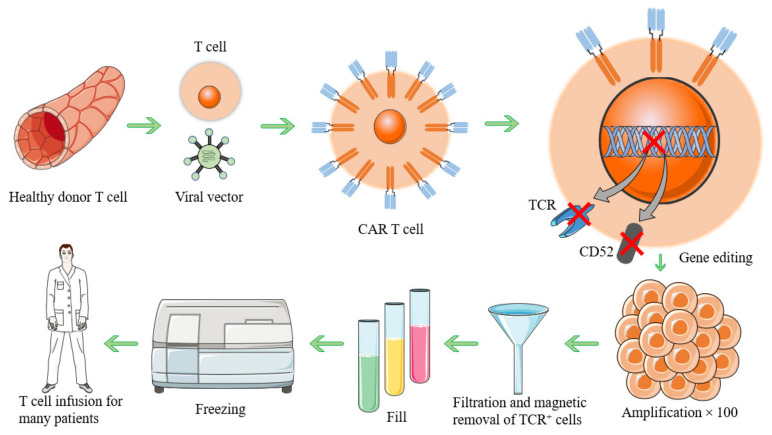
Flowchart of allogeneic CAR-T-cell production.

**Figure 2 biomolecules-13-00465-f002:**
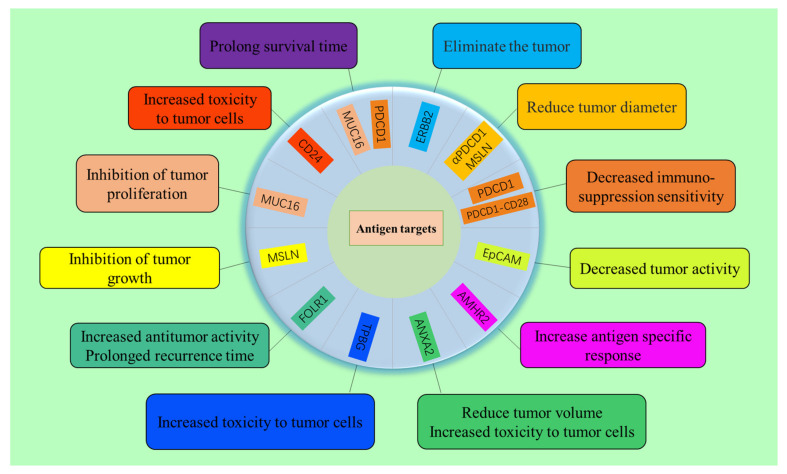
Antigenic targets of OC cells and related functions of CAR-T cells targeting these tumor antigens.

**Figure 3 biomolecules-13-00465-f003:**
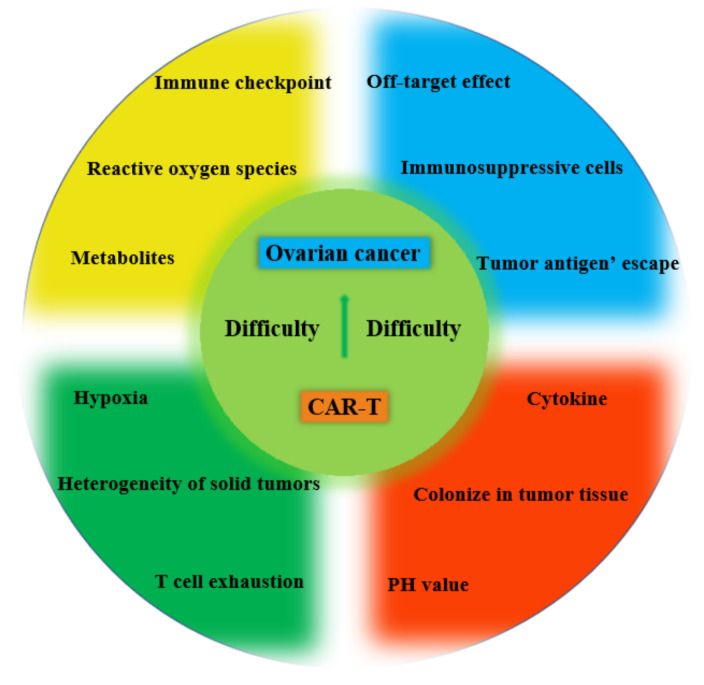
The challenges for CAR-T-cell therapy for the treatment of OC.

**Figure 4 biomolecules-13-00465-f004:**
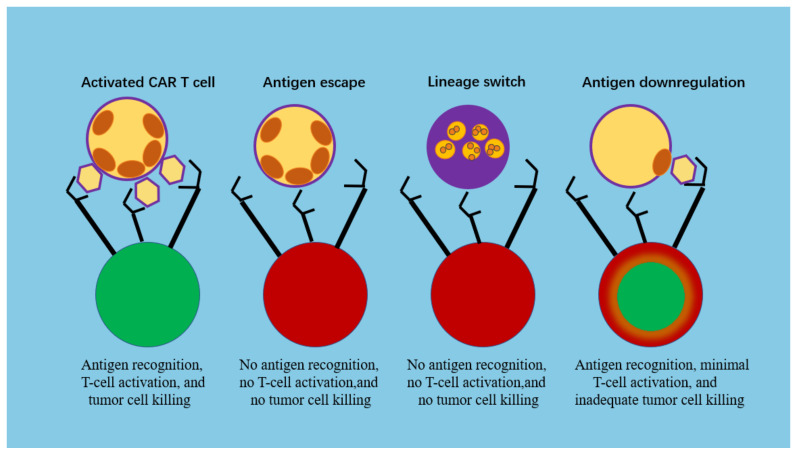
Mechanisms of tumor immune escape.

**Figure 5 biomolecules-13-00465-f005:**
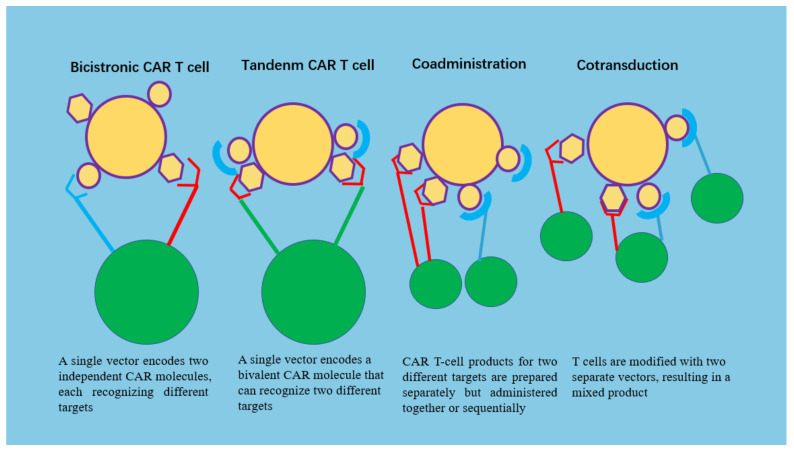
Different types of CAR-T cells are designed to inhibit immune escape.

**Figure 6 biomolecules-13-00465-f006:**
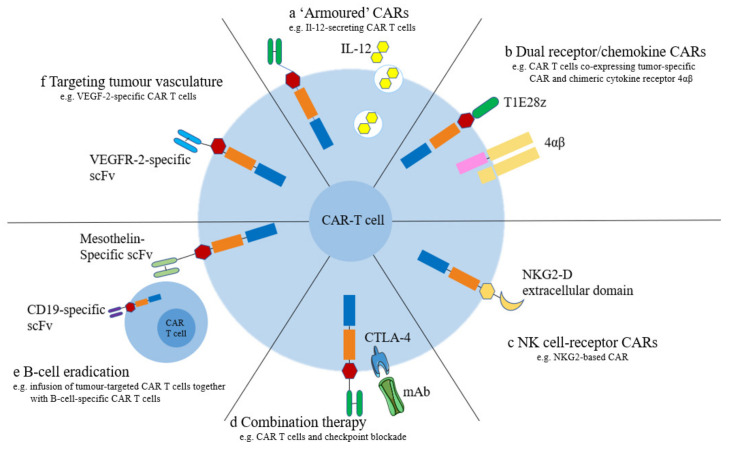
Schematic diagram of six methods to improve the therapeutic effect of CAR-T-cell therapy.

**Figure 7 biomolecules-13-00465-f007:**
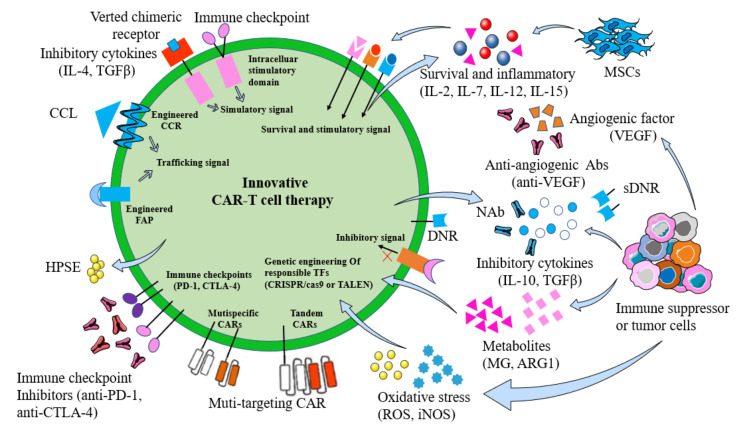
Ways to improve CAR-T-cell therapy.

**Figure 8 biomolecules-13-00465-f008:**
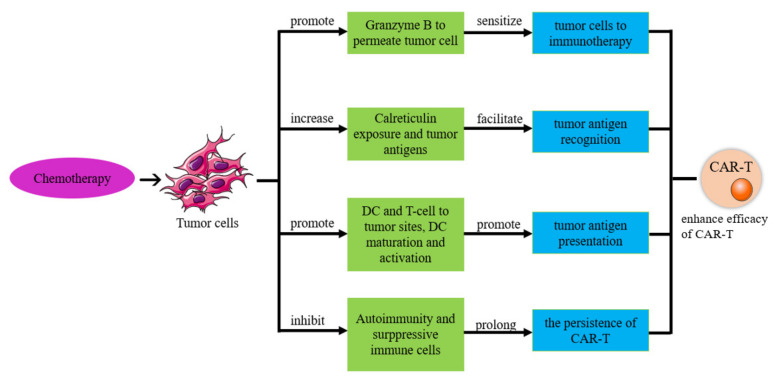
Mechanism of chemotherapy to improve the effect of CAR-T therapy.

**Figure 9 biomolecules-13-00465-f009:**
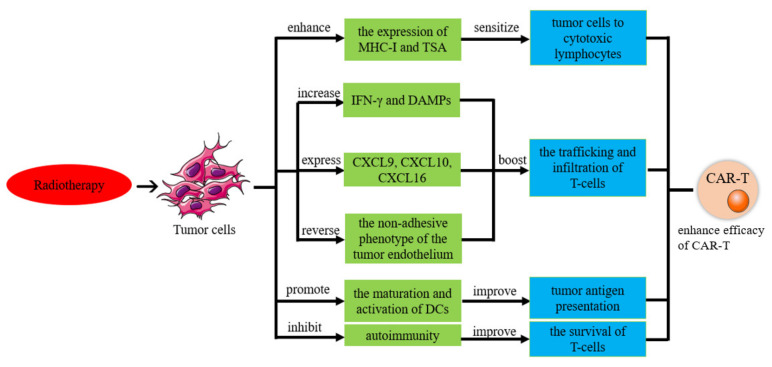
Mechanism diagram of radiotherapy to enhance the therapeutic effect of CAR-T-cell therapy.

## Data Availability

Not applicable.

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
