# Peer review of "CAR-T Cells in the Treatment of Ovarian Cancer: A Promising Cell Therapy"

_biomolecules, 2023, doi:10.3390/biom13030465_

Round 1

Reviewer 1 Report

The manuscript by Zhang et al., entitled “CAR-T Cell Receptor in the Treatment of Ovarian Cancer: A Promising Cell Therapy reviews the recent updates in CAR-T cell engineering for enhanced efficacy in ovarian cancer. This is timely review covering a very interesting subject. This review covers almost all aspects of CAR-T cells including their potential application in cancer therapy including various strategies to improve the efficacy.

Some of the specific concerns are given below:

1.     Authors should verify the cancer statistics as per latest update by WHO and IARC. In the abstract authors state “5-year survival rate of approximately 47%” whereas in the Introduction it is “5-year survival rate of less than 25%”. This is difficult to correlate and misleading. Also, please provide valid references.

2.     Abbreviations should be carefully used. Exact full forms of all abbreviations should be mentioned at the first instance and

3.     Line 83: “C Compared to TCR, CAR 83 antibodies can….” . I wonder what does the “C” stands for here.

4.     Line 119-120: “T-cell redirected for universal cytokine killing (LORRY)”. The abbreviation should be TRUCKs.

5.     Please check the fonts in abbreviations. For instance, Line 219 “cars” should be “CARs”.

6.     There are several such grammatical, typographical and language issues in the manuscript. Therefore, thorough language check is needed.

Author Response

Journal Editorial Office

Dear Editor,

We would like to resubmit the revised manuscript entitled “CAR-T Cells in the Treatment of Ovarian Cancer: A Promising Cell Therapy” for consideration by Biomolecules.

We would like to thank the reviewers for thoroughly reviewing our manuscript and making many thoughtful comments. We were very pleased to see that the reviewers recognized the novelty and potential significance of our work. We have revised the manuscript to address the reviewers’ comments. The manuscript has been edited by a professional language company recommended by the journal. The changes in the revised manuscript have been highlighted in red. Thank you for your consideration of our manuscript.

Yours sincerely,

Man-Hua Cui, MD, Department of gynecology, The Second Hospital of Jilin University, Changchun, Jilin Province, China; E-mail addresses: manhua@jlu.edu.cn.

Response to Reviewer 1 Comments

Reviewer 1

The manuscript by Zhang et al., entitled “CAR-T Cell Receptor in the Treatment of Ovarian Cancer: A Promising Cell Therapy” reviews the recent updates in CAR-T cell engineering for enhanced efficacy in ovarian cancer. This is timely review covering a very interesting subject. This review covers almost all aspects of CAR-T cells including their potential application in cancer therapy including various strategies to improve the efficacy.

Some of the specific concerns are given below:

1.Authors should verify the cancer statistics as per latest update by WHO and IARC. In the abstract authors state “5-year survival rate of approximately 47%” whereas in the Introduction it is “5-year survival rate of less than 25%”. This is difficult to correlate and misleading. Also, please provide valid references.

Response 1: Thank you very much for your professional and valuable comments. We have reviewed the literature and have revised the manuscript according to your comments. In addition, we cite the relevant reference1.

2.Abbreviations should be carefully used. Exact full forms of all abbreviations should be mentioned at the first instance and

Response 2: Thank you very much for your professional and valuable comments. We checked all the abbreviations and mentioned at the first instance.

3.Line 83: “C Compared to TCR, CAR 83 antibodies can….” . I wonder what does the “C” stands for here.

Response 3: Thank you very much for your professional and valuable comments. We have deleted the letter "C" from this sentence. Besides, the manuscript has been edited by a professional language company. We have uploaded the certificate.

4.Line 119-120: “T-cell redirected for universal cytokine killing (LORRY)”. The abbreviation should be TRUCKs.

Response 4: Thank you very much for your professional and valuable comments. We have revised the manuscript according to your opinion.

5.Please check the fonts in abbreviations. For instance, Line 219 “cars” should be “CARs”.

Response 5: Thank you very much for your professional and valuable comments. We have revised the manuscript according to your opinion.

6.There are several such grammatical, typographical and language issues in the manuscript. Therefore, thorough language check is needed.

Response 6: Thank you very much for your professional and valuable comments. We have revised the manuscript according to your opinion. We have deleted unimportant and repetitive paragraphs to make the manuscript more concise, and changed the logical order of the article to increase the reader's interest. The manuscript was edited by a professional language service provider recommended by the journal. In addition, we uploaded the certificate.

References

1. International agency for research on cancer. Survival, incidence, and mortality over time. https://gcoiarcfr/survival/survmark/.

Reviewer 2 Report

The review manuscript by Zhang and colleagues provides an extensive overview of CAR T cell therapies, their success in the treatment of hematological malignancies, and challenges for the treatment of solid tumors with a particular focus on ovarian cancers.

As general remarks, I find this review to be too extensive and addressing topics that may not be directly related to the treatment of ovarian cancer using CAR T cell therapy as the title suggests. The extension of it resembles a book chapter rather than a review paper. I would advise shortening the review and considering carefully the topics to address that are directly related to CAR T cell therapy for ovarian cancer and its challenges and successes. Moreover, from a quick search in the literature, I´ve found a review paper also addressing such matter that has been published by Yan et al in 2019 (Onco Targets Ther). The mentioned review has not been cited by the authors and I would advise to do so and to address in this review manuscript novel topics and results that complement or have been reported by Yan et al in order to increase the impact and relevance of this review manuscript.

Punctuation and spelling mistakes have been identified throughout the entire manuscript. Also, a lot of sentences seem to be duplicated within even the same paragraph bringing a lot of confusion to the reader and incoherency to the text. Examples among many others are: “Therefore, there is a great need for new treatment strategies in these patients.”  line 29/30 and again line 33/34; “With the improved understanding of the molecular basis of immune recognition and immune regulation in cancer cells” line 31/32 and again line 34/35; “relative inefficiency caused mainly by several factors: The relative inefficiency of traditional CAR-T cells is mainly caused by several factors:” line 1579-1581. Please consider a careful and thorough revision of the English language and punctuation.

I would suggest a thoughtful revision of the manuscript to improve its relevance/impact on the field, better understanding, and ease the reading of the content for the readers.

As a last comment, I would like to ask the authors for the meaning or purpose of the word “receptor” in the tile after the words “CAR-T Cell” (title: “CAR-T Cell Receptor in the Treatment of Ovarian Cancer: A Promising Cell Therapy”)? Given that CAR already includes and refers to receptors on T cells, I find the use of this term again redundant. 

Author Response

Journal Editorial Office

Dear Editor,

We would like to resubmit the revised manuscript entitled “CAR-T Cells in the Treatment of Ovarian Cancer: A Promising Cell Therapy” for consideration by Biomolecules.

We would like to thank the reviewers for thoroughly reviewing our manuscript and making many thoughtful comments. We were very pleased to see that the reviewers recognized the novelty and potential significance of our work. We have revised the manuscript to address the reviewers’ comments. The manuscript has been edited by a professional language company recommended by the journal. The changes in the revised manuscript have been highlighted in red. Thank you for your consideration of our manuscript.

Yours sincerely,

Man-Hua Cui, MD, Department of gynecology, The Second Hospital of Jilin University, Changchun, Jilin Province, China; E-mail addresses: manhua@jlu.edu.cn.

Response to Reviewer 2 Comments

Reviewer 2

The review manuscript by Zhang and colleagues provides an extensive overview of CAR T cell therapies, their success in the treatment of hematological malignancies, and challenges for the treatment of solid tumors with a particular focus on ovarian cancers.

As general remarks, I find this review to be too extensive and addressing topics that may not be directly related to the treatment of ovarian cancer using CAR T cell therapy as the title suggests. The extension of it resembles a book chapter rather than a review paper. I would advise shortening the review and considering carefully the topics to address that are directly related to CAR T cell therapy for ovarian cancer and its challenges and successes. Moreover, from a quick search in the literature, I´ve found a review paper also addressing such matter that has been published by Yan et al in 2019 (Onco Targets Ther). The mentioned review has not been cited by the authors and I would advise to do so and to address in this review manuscript novel topics and results that complement or have been reported by Yan et al in order to increase the impact and relevance of this review manuscript.

Response 1: Thank you very much for your professional and valuable comments. We have carefully studied your recommended article and have incorporated the new ideas from this review into our manuscript1,2.

Punctuation and spelling mistakes have been identified throughout the entire manuscript. Also, a lot of sentences seem to be duplicated within even the same paragraph bringing a lot of confusion to the reader and incoherency to the text. Examples among many others are: “Therefore, there is a great need for new treatment strategies in these patients.”  line 29/30 and again line 33/34; “With the improved understanding of the molecular basis of immune recognition and immune regulation in cancer cells” line 31/32 and again line 34/35; “relative inefficiency caused mainly by several factors: The relative inefficiency of traditional CAR-T cells is mainly caused by several factors:” line 1579-1581. Please consider a careful and thorough revision of the English language and punctuation.

I would suggest a thoughtful revision of the manuscript to improve its relevance/impact on the field, better understanding, and ease the reading of the content for the readers.

Response 2: Thank you very much for your professional and valuable comments. We have revised the manuscript according to your opinion. We have deleted unimportant and repetitive paragraphs to make the manuscript more concise and changed the logical order of the article to increase the reader's interest. The manuscript was edited by a professional language service provider recommended by the journal. In addition, we uploaded the certificate.

As a last comment, I would like to ask the authors for the meaning or purpose of the word “receptor” in the tile after the words “CAR-T Cell” (title: “CAR-T Cell Receptor in the Treatment of Ovarian Cancer: A Promising Cell Therapy”)? Given that CAR already includes and refers to receptors on T cells, I find the use of this term again redundant.

Response 3: Thank you very much for your professional and valuable comments. We revised the title according to your opinion.

References

  1. Yan L, Liu B. Critical factors in chimeric antigen receptor-modified T-cell (CAR-T) therapy for solid tumors. OncoTargets and Therapy. 2018;Volume 12:193-204.
  2. Yan W, Hu H, Tang B. Advances Of Chimeric Antigen Receptor T Cell Therapy In Ovarian Cancer. OncoTargets and Therapy. 2019;Volume 12:8015-8022.

Round 2

Reviewer 2 Report

Dear authors,

thank you for taking my comments into consideration and for your efforts to improve the manuscript. 

Author Response

Response to Reviewer 2 Comments

  1. General comment: The reviewers have provided useful comments who helped the authors to improve their paper.

Although the paper remains very scholar and sometimes confusing owing to repeats, it can be published provided that the corrections below be made.

  1. HGNC genes names should be used throughout the manuscript (with the definition and aliases, the first time a gene is quoted):

CD24

CD28

CD52

CD247: CD3ZETA, CD3z, CD3ζ

CD274: PD-L1

PDCD1: PD-1, programmed cell death 1

TNFRSF9: tumor necrosis factor receptor superfamily 9 (CD137, 4-1BB)

CCR2: C-C motif chemokine receptor 2

CCR4: C-C motif chemokine receptor 4

IL2: interleukin 2

IFNG: interferon gamma IFN-γ)

TNF: tumor necrosis factor (TNF-α)

TGFB : transforming growth factor beta (TGFβ)

ERBB2: erb-b2 receptor tyrosine kinase 2 (HER2)

MSLN: mesothelin (instead of MESO)

MUC16: mucin 16, cell surface associated

EPCAM: epithelial cell adhesion molecule (Ep-CAM)

FOLR1: folate receptor alpha (FRα)

AMHR2: anti-Müllerian hormone receptor type 2, a member of the TGFB receptor family (MISRII

Müllerian inhibiting substance type 2)

ANXA2: annexin A2

TPBG: trophoblast glycoprotein (5T4)

RUNX3: RUNX family transcription factor 3

NR4A1: nuclear receptor subfamily 4 group A number 1

CTLA4: cytotoxic T-lymphocyte associated protein 4

CD80: B7-1

CD86: B7-2

  1. Other comments:

Line 294: ‘Binding of PD-1/PD-L1/2’ > ‘PDCD1 following binding by its ligands CD274 and PDCD1LG2’

Line 296: ‘CTLA-4 binds to B7’ > ‘CTLA4 following binding by its ligands CD80 (B7-1) and CD86 (B7-2)’

Line 332, 349, 353: TEM > TME (check if others)

Line 421: TRG > Treg

Line 41: TCRs >TR (when referring to T cell receptors dimers only (i.e. without the CD3 coreceptors), TR should be used)

Line 42: delete ‘complex’ for major histocompatibility (MH) (when referring to genes and proteins MH should be used , MHC refers to the locus)

Line 43: MHC > MH

Line 45, 47, 48 …: TCR >TR

Line 60 : ‘T cell signaling region’ > ‘T cell coreceptors signaling region’

Line149: ‘EpCAM-scFv. The fragment targets’ > ‘EPCAM-scFv, the fragment that targets’

Line 189: ‘OC cell lines’ > ‘ovary cancer (OC) cell lines’

Line 219 : not clear. Seems a repeat. ‘Klapdor’s novel anti-CD24-CAR’ > ‘This dual CAR’ (exact ?)

  1. Figure 2 should be revised to be more scientifically driven (HGNC gene names) and more accurate in its content (for example, to represent scFv with a disulfide bridge between the V domains is misleading).

Response: We would like to thank you for thoroughly reviewing our manuscript and making many professional comments. We were very pleased to see that the reviewers recognized the novelty and potential significance of our work. We have revised the manuscript according to your comments. The changes in the revised manuscript have been highlighted in red. Thank you for your consideration of our manuscript.